# Equal K Amounts to N Achieved Optimal Biomass and Better Fiber Quality of Late Sown Cotton in Yangtze River Valley

**Xiaolei Ma, Saif Ali**[ID]**, Abdul Hafeez, Anda Liu, Jiahao Liu, Zhao Zhang, Dan Luo, Adnan Noor Shah**[ID] **and Guozheng Yang \***

MOA Key Laboratory of Crop Eco-physiology and Farming system in the Middle Reaches of Yangtze River, College of Plant Science and Technology, Huazhong Agricultural University, Wuhan 430000, China; xiaoleima@webmail.hzau.edu.cn (X.M.); drsaifbhatti@webmail.hzau.edu.cn (S.A.); ahafeez1226@webmail.hzau.edu.cn (A.H.); lada199@163.com (A.L.); liujiahao@webmail.hzau.edu.cn (J.L.); zzhang@webmail.hzau.edu.cn (Z.Z.); ld17191091496@126.com (D.L.); ans.786@yahoo.com (A.N.S.)

\* Correspondence: ygzh9999@mail.hzau.edu.cn; Tel.: +139-9555-3884

**Abstract:** Potassium (K) fertilizer plays a crucial role in the formation of the biological and economic yield of cotton (*Gossypium hirsutum* L.). Here we investigated the effects of the amount of K on biomass accumulation and cotton fiber quality with lowered N amounts (210 kg ha$^{-1}$) under late sowing, high density and fertilization once at 2 weeks after squaring. A 2-year field experiment was performed with three K fertilizer amounts (168 kg ha$^{-1}$ ($K_1$), 210 kg ha$^{-1}$ ($K_2$), and 252 kg ha$^{-1}$ ($K_3$)) using a randomized complete block design in 2016 and 2017. The results showed correspondingly, $K_3$ accumulated cotton plant biomass of 7913.0 kg ha$^{-1}$, next to $K_2$ (7384.9 kg ha$^{-1}$) but followed by $K_1$ (6985.1 kg ha$^{-1}$) averaged across two growing seasons. Higher K amounts ($K_2$, $K_3$) increased biomass primarily due to a higher accumulation rate (32.68%–74.02% higher than $K_1$) during the fast accumulation period (FAP). Cotton fiber length, micronaire, and fiber strength in $K_2$ were as well as $K_3$ and significantly better than $K_1$. These results suggest that K fertilizer of 210 kg ha$^{-1}$ should be optimal to obtain a promising benefit both in cotton biomass and fiber quality and profit for the new cotton planting model in the Yangtze River Valley, China and similar climate regions.

**Keywords:** cotton; potassium; fertilizer; biomass accumulation; fiber quality

## 1. Introduction

Cotton is one of the most important fiber crops grown not only for fiber but also for the paper and oil industries [1,2]. China is one of the leading countries for cotton production. The Yangtze River Valley is one of the three cotton-growing regions in China where seedlings are transplanted after wheat or rapeseed is harvested and more than 300 kg ha$^{-1}$ N is applied in three splits (30% at pre-plant, 40% at first bloom, and 30% at peak bloom) [3,4]. However, the arduous procedure and excess fertilizer input are depleting cotton production profits [5]. To improve production benefit, a new planting model with late sowing (mid-May) [6], high density (9–10 plants m$^{-2}$) [6,7], low N amounts (180–225 kg ha$^{-1}$) [7], and once fertilization [3,8] has been practiced as an effective way to fight the challenge of high cost in cotton production in the region. The new planting model harvested similar yield to the conventional practice [9] but greatly reduced the cost resulted from less manual work, low N fertilizer amount and less application of chemicals, due to the short cotton growing season with high planting density.

Numerous studies have demonstrated that K is a fundamental element for plant growth which markedly affects biomass accumulation and biomass partitioning [10–12]. Applying potassium fertilizer improved cotton plant biomass [13], especially the biomass of cotton bolls [14], and it increased the

reproductive parts biomass per unit area [15–17]. On the contrary, K deficiency reduced not only the production but also the transportation of dry matter, leading to poor growth and reduced biomass accumulation in bolls [18]. Excessive K fertilizer has increased not only luxurious consumption and environmental concern [19] but also canopy closure, leading to rotten bolls and delayed maturation [20]. Tsialtas et al. [21] revealed that 80 kg $K_2O$ ha$^{-1}$ was sufficient for cotton growth to achieve considerable yields in Australia. However, it remains to study how much K has to be applied to ensure enough cotton products for the new planting model. Previous studies have indicated that the cotton plant could produce a considerable yield of 2691 kg ha$^{-1}$ seed cotton when the K amount was in line with N amount. It is hypothesized that K could also be reduced in accordance with N because the plant should keep in balance in nutrients accumulation for normal growth and fruits.

Cotton fiber quality is an important standard in cotton production based on high yield. Many studies focused on the effect of K on cotton fiber quality traits but the results had many differences. Some studies showed that the K amount significantly affected the fiber length [21,22], strength, micronaire, uniformity, and elongation of the cotton [23]. However, some studies indicated that fiber properties were not significantly affected by the K amount [16,24,25].

The study aimed to (1) determine the effects of K fertilizer amount (ranging from 168–252 kg ha$^{-1}$ $K_2O$) on cotton phenology, biomass accumulation (duration and rate of FAP and distribution) and fiber quality; (2) find the optimal K amount to achieve high productivity and fiber quality of cotton in the new planting model.

## 2. Materials and Methods

### 2.1. Experimental Site and Cultivar

The field experiment was conducted in 2016 and 2017 with Huamian 3109 (*G. hirsutum* L.) on the experimental farm of Huazhong Agricultural University, Wuhan, China (30°37′ N latitude, 114°21′ E longitude, 23 m elevation). The soil of the experimental field was yellowish-brown and clay loam comprising of 89.3 mg kg$^{-1}$ alkaline N, 26.4 mg kg$^{-1}$ $P_2O_5$, and 177.0 mg kg$^{-1}$ $K_2O$.

### 2.2. Climate

The mean air temperatures from May to October in 2016 were 25.4 °C with 0.1 °C lower than that in 2017, and from June to September, air temperatures in 2016 were 0.3–1.6 °C lower than that in early 2017. The total rainfall from May to October in 2016 was 1311.6 mm with 925 mm more than that in 2017, and rainfall was mainly concentrated on June and July in 2016 with 823 mm more than that in 2017, but 107 mm less from August to September in 2016 than in 2017 [26].

### 2.3. Experiment Design

A randomized complete block design was employed with four replicates. Three K fertilizer amounts were 168 kg ha$^{-1}$ ($K_1$), 210 kg ha$^{-1}$ ($K_2$), and 252 kg ha$^{-1}$ ($K_3$).

Fertilizers, as provided by urea (46.3% N) for 210 kg N ha$^{-1}$, calcium superphosphate (12% $P_2O_5$) for 63 kg $P_2O_5$ ha$^{-1}$, potassium chloride (59% $K_2O$) for three amounts, and borate (10% B) for 1.5 kg B ha$^{-1}$, were mixed evenly and buried in 10 cm deep between cotton rows in bed 2 weeks after squaring.

### 2.4. Field Management

The plant density was $9 \times 10^4$ plants ha$^{-1}$ with a row to row space of 76 cm. The plot size was 36.48 m$^2$ (12 m × 3.04 m) with four rows in two beds. Cotton seeds were sown directly on 18 May 2016 and 10 May 2017. Seedlings were thinned at the three leave stage to the target planting density. Other field managements were carried out according to conventional practice.

*2.5. Data Collection*

2.5.1. Cotton Phenology

Fifteen successive and uniform plants in one row from each plot were fixed for the investigation of plant growth stages, such as squaring (50% plant bearing squares), first bloom (50% plants showing flowers), peak bloom (normally 15 d after first bloom), boll opening (50% plants showing open boll), and plant senescence. The specific growth period in days were identified as the duration from the day of the first stage to the day of the next stage, such as seedling, from emergence to squaring; squaring, from squaring to first bloom; flowering, from first bloom to peak bloom; boll setting, from peak bloom to boll opening; flowering and boll setting, from first bloom to boll opening; boll opening (or maturation), from boll opening to plant senescence.

2.5.2. Cotton Biomass Accumulation

Cotton biomass was measured five times (squaring, first bloom, peak bloom, boll opening, and plant senescence) in the fourth replication. Nine (eighteen at squaring stage) successive plants were carefully uprooted and grouped randomly but equally in number into 3 as replicates from each plot at each stage. Plants were separated into vegetative parts (root, stem, and leaves) and reproductive parts (square, flower, and boll). Sub-samples were packed separately and dried in an electric fan-assisted oven at 105 °C for 30 min, at 80 °C for constant weight, and then weighted. Vegetative part biomass (VPB) is the total biomass of root, stem, and leaves, and reproductive part biomass (RPB) is the total biomass of squares, flowers, and bolls, and cotton plant biomass (CPB) is the sum of VPB and RPB.

Cotton plant biomass accumulation progress was described by a logistic regression model [3],

$$W = \frac{W_M}{1 + a\mathrm{e}^{bt}},\tag{1}$$

where $a$ and $b$ are constants to be found, $t$ is the time as the days after emergence (DAE), $W$ is the biomass (g) at $t$, and $W_M$ is the maximum biomass (g).

According to Equation (1), the following equations will be calculated:

$$t_1 = \frac{1}{b}\ln\left(\frac{2 + \sqrt{3}}{a}\right),\tag{2}$$

$$t_2 = \frac{1}{b}\ln\left(\frac{2 - \sqrt{3}}{a}\right),\tag{3}$$

$$T = -\frac{\ln a}{b},\tag{4}$$

$$V_T = \frac{W_1 - W_2}{t_1 - t_2},\tag{5}$$

$$V_M = -\frac{bW_M}{4},\tag{6}$$

where $t_1$ and $t_2$ (DAE) are the initiation and termination of FAP (fast accumulation period), respectively; $T$ (d) is the duration of FAP; $V_T$ and $V_M$ (g d$^{-1}$) are the average and the highest biomass accumulation rate during FAP, respectively; $W_1$ and $W_2$ are the biomass at $t_1$ and $t_2$, respectively.

The accumulation rate (AR) of cotton plant biomass during each period was calculated by the following formula:

$$AR\left(\mathrm{kg\ ha^{-1}d^{-1}}\right) = \frac{W_T - W_I}{\text{period length}},\tag{7}$$

where $W_I$ and $W_T$ (kg ha$^{-1}$) are the biomasses on the first day and the last day of the period, respectively, and the period length (d) is the duration in days of this period.

### 2.5.3. Cotton Fiber Quality

One hundred maturated bolls were picked from each plot before harvest to get the fiber samples. High volume instrumentation (HVI) was used to analyze fiber quality parameters for each fiber sample, as described by [15]. The reports of five important quality parameters describing the fiber length, strength, fineness, elongation, uniformity was provided by HVI.

### 2.6. Statistical Analysis

Data are processed with Microsoft Excel 2010; ANOVA was performed with SPSS 21.0 (IBM Company, Chicago, IL, USA) and figures were drawn with Sigma Plot 12.5 (Systat Software Inc., San Jose, CA, USA). Least Significant Difference (LSD) among the treatments was conducted with Duncan at a 5% probability level ($p = 0.05$).

Higher K fertilizer amounts ($K_2$ and $K_3$) increased 10.34%–20.03% seed cotton yield over $K_1$ due to higher boll density in 2016 and boll weight in both years, although differences existed between years in yield and its components [26,27].

## 3. Results

### 3.1. Cotton Plant Phenology

Cotton flowering and boll setting period took the longest while squaring the shortest, although differences existed between years in each specific cotton growth period (seedling, squaring and flowering, and boll setting) (Table 1).

**Table 1.** Cotton growth stages and periods influenced by K fertilizer amounts.

| Year | Treatment | Growing Stage (m/d) # | | | | Growth Period (d) # | | | |
|------|-----------|----------|----------|-------------|---------|----------|----------|----------------------------|-------|
| | | Emergence | Squaring | First Bloom * | Opening | Seedling | Squaring | Flowering and Boll Setting | Total |
| 2016 | $K_1$ | 5/28 | 7/15 | 8/1 | 9/23 | 48a * | 17a | 53a | 118a |
| | $K_2$ | 5/28 | 7/15 | 8/1 | 9/22 | 48a | 17a | 52a | 117a |
| | $K_3$ | 5/28 | 7/15 | 8/1 | 9/22 | 48a | 17a | 52a | 117a |
| 2017 | $K_1$ | 5/18 | 6/20 | 7/15 | 8/24 | 33a | 25a | 40a | 98a |
| | $K_2$ | 5/18 | 6/20 | 7/15 | 8/25 | 33a | 25a | 41a | 99a |
| | $K_3$ | 5/18 | 6/20 | 7/15 | 8/25 | 33a | 25a | 41a | 99a |

# m/d shows month/date, d means days. * Values followed by different letters within the same column in the same year are significantly different at probability levels ($p < 0.05$) according to the Least Significant Difference (LSD) test.

None of the specific cotton growth periods were affected by the K fertilizer amounts within the same year. However, the cotton growth period in 2016 was 18 d longer than that in 2017, due to 15 d longer in seedling and 11 d longer in boll setting, but 8 d shorter in squaring.

### 3.2. Cotton Plant Biomass Accumulation

Cotton plant biomass (CPB) was significantly increased with increased K amounts in both years (Table 2). The same trends were observed in root and stem biomass. Compared with $K_1$, $K_2$ increased root and stem 11.55% and 2.11% in 2016, respectively. However, the root biomass in $K_2$ was lower than that in $K_1$ in 2017 with no significant difference and stem biomass in $K_2$ was 13.87% higher than that in $K_1$. Cotton plants in $K_3$ produced 24.71% (2016) and 0.65% (2017) more root biomass and 27.36% (2016) and 26.40% (2017) more stem biomass compared with $K_1$. Leaves and reproductive parts accumulated higher in $K_2$ and $K_3$, and significantly lower in $K_1$. The ratios of RPB to CPB had no significant difference among the three K amounts in 2016, but that is significantly higher in $K_2$ and $K_3$ than $K_1$. There were no significant differences between $K_2$ and $K_3$ for Leaves and RPB and the ratios of RPB to CPB. Furthermore, the ratios of RPB/CPB in $K_2$ were higher than other treatments.

**Table 2.** Cotton and each part biomass accumulation influenced by K fertilizer amounts.

| Year | Treatment | Biomass Accumulation (kg ha$^{-1}$) | | | | | RPB/CPB (%) |
| --- | --- | --- | --- | --- | --- | --- | --- |
| | | Root | Stem | Leaves | Reproductive Parts | Total | |
| 2016 | $K_1$ | 815.6c * | 1950.1b | 406.0b | 3258.8a | 6603.9b | 48.35a |
| | $K_2$ | 909.8b | 1991.3b | 494.6a | 3338.6a | 6759.6b | 49.39a |
| | $K_3$ | 1017.1a | 2206.3a | 517.1a | 3464.0a | 7086.8a | 48.86a |
| | Average | 914.2 | 2049.2 | 499.6 | 3353.8 | 6816.8 | 48.87 |
| 2017 | $K_1$ | 1169.1a | 2374.0c | 406.0b | 3417.1b | 7366.2c | 44.18b |
| | $K_2$ | 1103.1a | 2703.3b | 494.6a | 3709.1ab | 8010.1b | 46.32a |
| | $K_3$ | 1176.7a | 3000.8a | 517.1a | 4044.6a | 8739.2a | 46.29a |
| | Average | 1149.6 | 2692.7 | 472.6 | 3723.6 | 8038.5 | 45.60 |

* Values followed by different letters within the same column in the same year are significantly different at probability level ($p < 0.05$) according to Least Significant Difference (LSD) test.

The growth curves of CPB, VPB, and RPB increased along with the cotton growth stage following a sigmoid curve with different slopes from K fertilizer amounts (Figure 1).

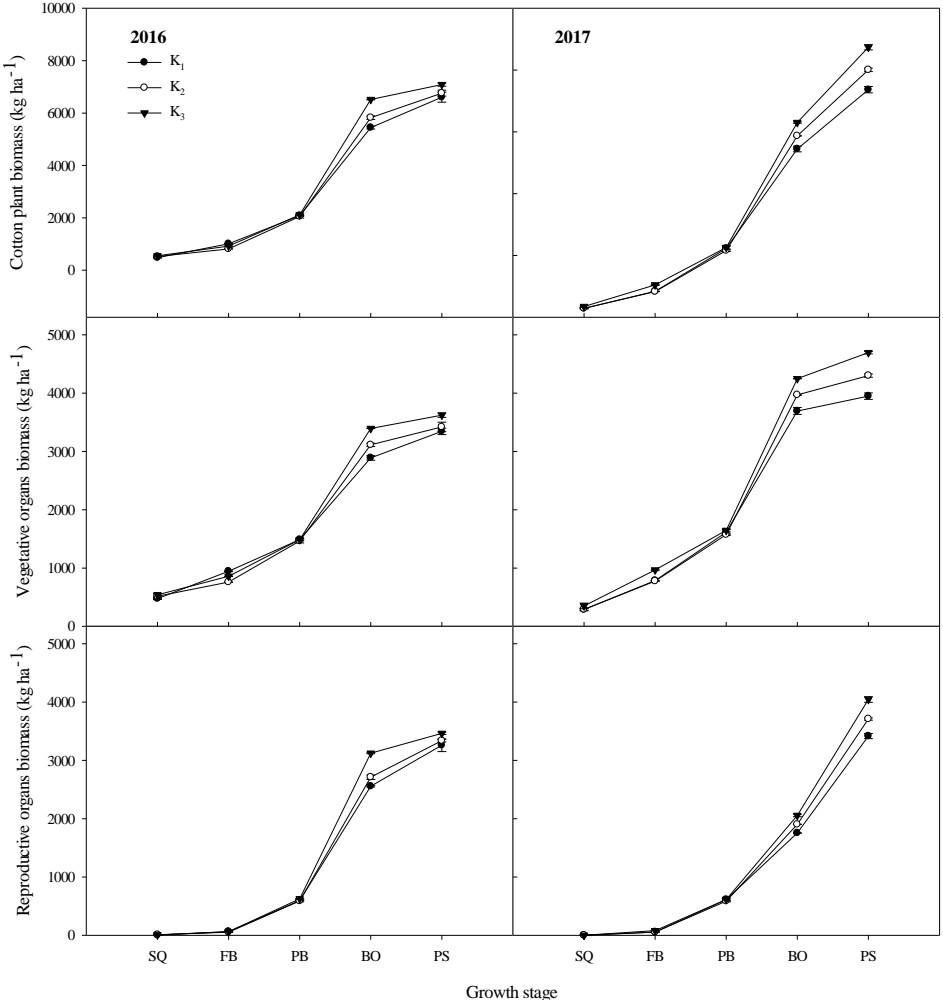

**Figure 1.** Cotton plant, vegetative parts, reproductive parts, and reproductive related parts biomass of field-grown cotton influenced by K fertilizer amounts at different growth stages in 2016. SQ, FB, PB, BO, and PS indicate squaring (51 days after emergence (DAE)), first bloom (66 DAE), peak bloom (81 DAE), boll opening (128 DAE), and plant senescence (168 DAE) stage, respectively. Error bar plus shows SEMs.

The growth curve slopes of CPB, VPB, and RPB were gradually increased until the boll opening stage and then decreased. Compared with RPB, the slopes of VPB curves were higher before peak bloom stage, and 41.04%–44.47% and 35.05%–40.90% VPB was produced in 2016 and 2017, respectively. However, RPB grew faster after peak bloom stage, and 81.75%–82.40% and 82.09%–84.82% RPB were produced in 2016 and 2017, respectively. The growth curves of CPB, VPB, and RPB in different K amounts gradually diverged from peak bloom. At the plant senescence stage, K3 plants produced 4.84% CPB, 5.90% VPB, and 3.76% RPB more than K2 plants and 7.31% CPB, 8.30% VPB, and 6.30% RRB more than K1 plants, respectively, in 2016. K3 plants produced 9.10% CPB, 9.15% VPB, and 9.04% RPB more than K2 plants and 18.64% CPB, 18.88% VPB, and 18.36% RRB more than K1 plants, respectively, in 2017.

### 3.3. Simulation of Biomass Accumulation

The biomass of cotton plants and each plant part accumulated following the logistic regression Equation (1) ($p < 0.01$) and showed different accumulation characteristics during FAP in both years (Table 3). Eigenvalues of cotton plant biomass accumulation were calculated by Equations (2)–(6) using the coefficient of Equation (1).

**Table 3.** Regression equation and Eigenvalues of cotton plant biomass accumulation of field-grown cotton influenced by K fertilizer amount in 2016 and 2017.

| Year | Treatment | Regression Eqs. $W =$ kg/ha, $t =$ DAE | $p$-Value | Fast Accumulation Period | | | | |
|------|-----------|-----------------------------------------|-----------|-----------------|-----------------|------------|--------------------|--------------------|
| | | | | $t_1$ (DAE) | $t_2$ (DAE) | $\Delta t$ | $VT$ (kg/ha d$^{-1}$) | $VM$ (kg/ha d$^{-1}$) |
| **Cotton plant** | | | | | | | | |
| 2016 | K1 | $W=6771.764/(1+4.98e^{-0.050t})$ | 0.0006 | 72.7 | 125.0 | 52.3 | 74.7 | 85.2 |
| | K2 | $W=6850.748/(1+5.58e^{-0.057t})$ | 0.0007 | 74.1 | 120.0 | 45.8 | 86.3 | 98.4 |
| | K3 | $W=7170.924/(1+6.19e^{-0.066t})$ | 0.0004 | 74.0 | 114.0 | 40.0 | 103.5 | 118.0 |
| | Average | | | 73.6 | 119.7 | 46.0 | 88.2 | 100.6 |
| 2017 | K1 | $W=7387.145/(1+6.18e^{-0.073t})$ | 0.0003 | 66.7 | 102.9 | 36.1 | 118.0 | 134.6 |
| | K2 | $W=8036.283/(1+6.54e^{-0.076t})$ | 0.0003 | 68.7 | 103.3 | 34.6 | 134.0 | 152.8 |
| | K3 | $W=8785.985/(1+6.33e^{-0.073t})$ | 0.0005 | 68.6 | 104.6 | 36.0 | 140.8 | 160.6 |
| | Average | | | 68.0 | 103.6 | 35.6 | 130.9 | 149.3 |
| **Vegetative parts** | | | | | | | | |
| 2016 | K1 | $W=3399.527/(1+4.00e^{-0.046t})$ | 0.0008 | 58.7 | 116.4 | 57.7 | 34.0 | 38.8 |
| | K2 | $W=3477.372/(1+4.61e^{-0.053t})$ | 0.0010 | 62.6 | 112.7 | 50.0 | 40.1 | 45.8 |
| | K3 | $W=3714.961/(1+4.71e^{-0.054t})$ | 0.0010 | 63.2 | 112.2 | 49.0 | 43.8 | 49.9 |
| | Average | | | 61.5 | 113.7 | 52.3 | 39.3 | 44.8 |
| 2017 | K1 | $W=4036.745/(1+6.64e^{-0.087t})$ | 0.0041 | 61.0 | 91.2 | 30.2 | 77.2 | 88.1 |
| | K2 | $W=4390.263/(1+7.01e^{-0.090t})$ | 0.0044 | 63.2 | 92.4 | 29.2 | 86.7 | 98.9 |
| | K3 | $W=4801.295/(1+6.43e^{-0.082t})$ | 0.0066 | 62.3 | 94.3 | 32.1 | 86.5 | 98.6 |
| | Average | | | 62.1 | 92.6 | 30.5 | 83.5 | 95.2 |
| **Reproductive parts** | | | | | | | | |
| 2016 | K1 | $W=3315.932/(1+6.97e^{-0.064t})$ | 0.0021 | 88.0 | 129.0 | 41.0 | 46.7 | 53.3 |
| | K2 | $W=3374.119/(1+7.34e^{-0.069t})$ | 0.0017 | 87.7 | 126.0 | 38.4 | 50.8 | 57.9 |
| | K3 | $W=3450.948/(1+8.52e^{-0.085t})$ | 0.0009 | 84.8 | 115.8 | 31.0 | 64.3 | 73.3 |
| | Average | | | 86.8 | 123.6 | 36.8 | 53.9 | 61.5 |
| 2017 | K1 | $W=3445.530/(1+6.85e^{-0.070t})$ | 0.0020 | 79.3 | 117.0 | 37.7 | 52.7 | 60.1 |
| | K2 | $W=3733.200/(1+7.25e^{-0.074t})$ | 0.0012 | 80.4 | 116.1 | 35.7 | 60.4 | 68.9 |
| | K3 | $W=4072.099/(1+7.27e^{-0.074t})$ | 0.0009 | 80.7 | 116.4 | 35.7 | 65.9 | 75.1 |
| | Average | | | 80.1 | 116.5 | 36.4 | 59.7 | 68.0 |

Where $t_1$ and $t_2$ (DAE) mean the initiation and termination, respectively, of the fast accumulation period (FAP); $\Delta t$ (d) means the duration of FAP; $V_T$ and $V_M$ (g d$^{-1}$) mean the average, and the highest biomass accumulation rate, respectively, during FAP.

The $W_M$ values of the logistic regression equation in CPB, VPB, and RPB were higher with increased K amounts (Table 2). The average and the highest biomass accumulation rates of CPB and each part biomass during FAP showed higher values in higher K amounts in both years.

Compared with RPB, VPB initiated FAP 25 d earlier in 2016 and 18 d in 2017 and terminated FAP 10 d earlier in 2016 and 24 d in 2017 with 15.5 d longer duration in 2016 and 6 d shorter in 2017. CPB initiated FAP 13 d (2016) and 12 d (2017) earlier than RPB and terminated 4 d (2016) and 13 d (2017) earlier with 9 d longer duration in 2016 and 1 d shorter in 2017. Compared with the average accumulation rates of FAP in RPB, the rates in CPB was 34.3 kg ha$^{-1}$ d$^{-1}$ (2016) and 71.2 kg ha$^{-1}$ d$^{-1}$ (2017) faster, and the rates in VPB was 14.6 kg ha$^{-1}$ d$^{-1}$ slower in 2016 and 23.8 kg ha$^{-1}$ d$^{-1}$ faster in 2017.

CPB initiated FAP in flowering and boll setting period (74 DAE in 2016 and 68 DAE in 2017) and terminated at 120 DAE (2016) and 104 DAE (2017) with the duration of 46 d and 36 d averaged across three K fertilizer amount in 2016 and 2017, respectively. The FAP initiations in $K_2$ and $K_3$ were similar but later than that in $K_1$ in both years. The FAP termination was earlier with increased K amounts in 2016, but later in 2017. The FAP duration was decreased with increased K amounts in 2016, but no similar result was observed in 2017.

VPB initiated FAP at 62 DAE in both years and terminated in flowering and boll setting period at 114 DAE (2016) and 93 DAE (2017), respectively. Many differences existed in FAP durations and biomass accumulation rates of VPB between both years which reflected the VPB were accumulated more slowly in 2016 than in 2017. With the increase in K amount, the duration of FAP become shorter and the accumulation rates were higher in 2016, but the shortest duration and highest accumulation of FAP in VPB were observed in $K_2$ in 2017.

RPB initiated and terminated FAP in the flowering and boll setting period with 37 d FAP duration in both years. The accumulation rates were higher in 2017 than in 2016. With increased K amounts, FAP durations decreased and the accumulation rates increased.

*3.4. Fiber Quality*

K amount significantly affected the fiber length, micronaire, and fiber strength. With increased K amount, the fiber length and the fiber strength increased significantly but there was no significant difference between that in $K_2$ and $K_3$. The micronaire values in different K amounts were no significant difference in 2016, but significantly lower in $K_1$ in 2017 with no significant difference between that in $K_2$ and $K_3$.

## 4. Discussion

K fertilizer is one of the main cotton fertilizers and has great correlations with cotton growth and the economic benefits of cotton production.

Many studies also reported that K deficiency could accelerate the growth process of cotton and result in premature senescence [28–31]. However, another study showed that K deficiency elicited similar effects on cotton earliness with late sowing which delayed flowering and boll development [32]. In the present study, the growth period was not affected by K amount, although apparent differences existed between the two growing seasons (Table 1). This might be due to the closeness of the K amount range in this study which was not sufficient to bring significant differences in growth period among different K amounts in the same year. Furthermore, the K amounts of three treatments were within the appropriate range for cotton growth under medium fertility, ensuring no K-deficiency in this study. The big differences between the two growing seasons were possibly due to a large amount of precipitation during the early cotton growth period but draught occurred in the flowering and boll setting periods in the 2016 growing season.

The biological yield was the basis of economic yield. Biomass accumulation could be explained in the context of plant photosynthesis, photo-assimilate translocation from vegetative to reproductive parts. K fertilizer affected the photo-assimilate export from leaves to sink parts and regulated the

sugar signaling in reproductive parts [24]. Potassium deficiency led to a reduction of main stem length, nodes and bolls, and also leaf photosynthesis and stomatal conductance [18,23,33,34], resulting in less carbohydrate production, a small sink and an in-balanced source-sink ratio. A previous study also revealed that a linear effect between K amounts and the growth efficiency of the reproductive part [35]. In this study, the RPB and the ratios of RPB to CPB were significantly higher in $K_2$ and $K_3$ with no significant difference between the two treatments (Table 2). This indicated that $K_2$ can benefit the carbohydrate production transportation from vegetative parts to reproductive parts and get the approximate RPB with $K_3$. Higher carbohydrate production in the reproductive parts can result in higher yield [36]. In the present study, the biomass of each cotton part was increased along with the increase in K amount (Table 2 and Figure 1). Similar studies revealed that increasing the K amount can increase the biomass of total plant and cotton bolls [14,18,37]. Furthermore, in this study, the FAP of CPB and RPB were initiated in the flowering and boll setting period and the FAP durations of RPB in both years were the same but the FAP accumulation rates were higher in 2017 with higher RPB (Table 3). With the increased K amount, the FAP accumulation rates were higher with higher biomass accumulation, but the duration of FAP shortened (Table 3). Similar results were also observed in Khan et al. [6] and Tung et al. [38]. This indicated that higher K amounts increased the cotton plant biomass mainly by higher accumulation rate during flowering and boll setting period.

In this study, fiber length, fiber strength, and micronaire were significantly affected by K amounts and better fiber quality traits were observed in $K_2$ and $K_3$ with no significant difference between $K_2$ and $K_3$ (Table 4). That indicated the fiber quality in $K_2$ was as well as that in $K_3$ and significantly better than that in $K_1$.

**Table 4.** Cotton fiber quality influenced by K fertilizer amounts.

| Year | Treatment | Length (mm) | Uniformity (%) | Micronaire | Strength (g/tex) | Elongation (%) |
|------|-----------|-------------|----------------|------------|------------------|----------------|
| 2016 | $K_1$ | 23.3b * | 83.8a | 4.6a | 25.8b | 6.57a |
|      | $K_2$ | 25.0ab | 83.8a | 4.8a | 27.6ab | 6.60a |
|      | $K_3$ | 25.4a | 84.1a | 4.5a | 28.7a | 6.60a |
|      | Average | 24.6 | 83.9 | 4.6 | 27.4 | 6.6 |
| 2017 | $K_1$ | 22.2b | 84.6a | 5.2a | 24.6b | 6.57a |
|      | $K_2$ | 23.1a | 85.1a | 4.4b | 26.0a | 6.60a |
|      | $K_3$ | 23.6a | 84.9a | 4.3b | 26.4a | 6.60a |
|      | Average | 23.0 | 84.9 | 4.6 | 25.7 | 6.6 |

* Values followed by different letters within the same column in the same year are significantly different at probability level ($p < 0.05$) according to Least Significant Difference (LSD) test.

## 5. Conclusions

K amounts ranging from 168–252 kg $K_2O$ ha$^{-1}$ have not altered the cotton growth period. Higher K increased cotton biomass due to a higher accumulation rate during FAP. Nevertheless, $K_2$ had similar fiber qualities, biomass accumulation, and partitioning as $K_3$.

The results suggest that an equal K amount to lowed N of 210 kg ha$^{-1}$ should be the optimal strategy under this new planting model in Yangtze River Valley, China, and similar regions.

**Author Contributions:** Conceptualization, X.M. and G.Y.; methodology, X.M.; data curation, X.M., S.A., A.H., A.L., J.L., Z.Z., D.L., A.N.S. and G.Y.; writing—original draft preparation, X.M.; writing—review and editing, X.M. and G.Y.; funding acquisition, G.Y. All authors have read and agreed to the published version of the manuscript.

**Funding:** This research was funded by the National Natural Science Foundation of China (31271665, 31771708).

**Conflicts of Interest:** The authors declare no conflict of interest.

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
