# Peer review of "Equal K Amounts to N Achieved Optimal Biomass and Better Fiber Quality of Late Sown Cotton in Yangtze River Valley"

_agronomy, doi:10.3390/agronomy10010112_

Round 1
Reviewer 1 Report
This manuscript presents the effect of K application on cotton biomass production and discusses about the relationship between biomass production and cotton yield. These studies are important for the cotton production in Yangtze River Valley, China and similar climate regions. Although the paper provides interesting data but it still needs a revision to be acceptable for the journal.
The accumulation rates derived from the measured values are mentioned in the section 3.2 Cotton plant biomass accumulation, and those derived from the model are mentioned in the section 3.4. Dynamic of biomass accumulation rate. It would be useful to combine these two sections to describe results of the accumulation rates. In addition, very few results derived from the model analysis are discussed in Discussion. These suggests that the model analysis is not necessary to lead the conclusion. More explanations on the model analysis should be given in Discussion or delete the model analysis.
Specific comments
Line 63; The exact mean of air temperatures and total rainfall in addition to the differences among years should be indicated.
Line 90; Methodology for cotton yield and yield components should be removed since data of cotton yield are referred.
Lines 99-102; Abbreviations such as VPB would be mentioned in this sentence.
Line 127; The data of cotton yield and yield components would be mentioned in Introduction with enough information such as exact measured cotton yield values, to understand the effect of K application on cotton yield.
Line 195; “The accumulation speeds of CPB during FAP were higher than that of each plant parts”, but this result is rather predictable since sum of accumulation speeds of each plant parts is the accumulation speeds of CPB.
Lines 287-288; Plants are separated differently in 2016 and 2017 except for RPB in 2016 and SNB in 2017. Thus, comparing SPB and FPB with VPB and RRB may not be suitable.
Lines 307-321; Part of these two sections are not be suitable for the present study since the effect of K application on cotton yield and yield components, which are referred, are discussed.
Figure 1; It is very difficult to recognize the accumulation rate mentioned in 3.2. Cotton plant biomass accumulation from the Figure 1. The value of x-axis would be changed to the days after emergence.
Table 4; Statistical analysis for correlation should be mentioned in Materials and Method, especially for how to deal replications.
Author Response
Response to Reviewer 1 Comments
Point 1: The accumulation rates derived from the measured values are mentioned in the section 3.2 Cotton plant biomass accumulation, and those derived from the model are mentioned in the section 3.4. Dynamic of biomass accumulation rate. It would be useful to combine these two sections to describe results of the accumulation rates. In addition, very few results derived from the model analysis are discussed in Discussion. These suggests that the model analysis is not necessary to lead the conclusion. More explanations on the model analysis should be given in Discussion or delete the model analysis.
Response 1: I had followed your suggestion, and deleted the part of “Dynamic of biomass accumulation rate” and had added more explanations on the model analysis in Discussion.
Point 2: Line 63; The exact mean of air temperatures and total rainfall in addition to the differences among years should be indicated.
Response 2: I had followed your suggestion, and add the exact mean of air temperatures and total rainfall.
Point 3: Line 90; Methodology for cotton yield and yield components should be removed since data of cotton yield are referred.
Response 3: I had followed your suggestion, and removed it.
Point 4: Lines 99-102; Abbreviations such as VPB would be mentioned in this sentence.
Response 4: I had followed your suggestion, and mentioned in this sentence.
Point 5: Line 127; The data of cotton yield and yield components would be mentioned in Introduction with enough information such as exact measured cotton yield values, to understand the effect of K application on cotton yield.
Response 5: I had followed your suggestion, and mentioned it.
Point 6: Line 195; “The accumulation speeds of CPB during FAP were higher than that of each plant parts”, but this result is rather predictable since sum of accumulation speeds of each plant parts is the accumulation speeds of CPB.
Response 6: I had followed your suggestion, and deleted it.
Point 7: Lines 287-288; Plants are separated differently in 2016 and 2017 except for RPB in 2016 and SNB in 2017. Thus, comparing SPB and FPB with VPB and RRB may not be suitable.
Response 7: I had followed your suggestion, and deleted it.
Point 8: Lines 307-321; Part of these two sections are not be suitable for the present study since the effect of K application on cotton yield and yield components, which are referred, are discussed.
Response 8: I had followed your suggestion, and deleted it.
Point 9: Figure 1; It is very difficult to recognize the accumulation rate mentioned in 3.2. Cotton plant biomass accumulation from the Figure 1. The value of x-axis would be changed to the days after emergence.
Response 9: I need to explain this point for you that the plant sample were taken according to plant growth stage and every growth stage was different days after emergency in two years, so I haven’t change the x-axis into DAE, but put the two year biomass accumulation figures in one figure.
Point 10: Table 4; Statistical analysis for correlation should be mentioned in Materials and Method, especially for how to deal replications.
Response 10: I had followed your suggestion, and I had deleted this part.
Reviewer 2 Report
This is a good contribution to the science of K application in cotton. Below are my detailed comments:
Title: It is not clear what "equal K" means. Do you mean K amounts equal to that of N application?
Page 1, Lines 13-14 - Replace "…a lowed N amounts (210 kg ha-1) under a condition of late sowing,…” with “… low N amounts (210 kg ha-1) under late sowing…”
Page 1, Line 20 (and other parts of the manuscript) – Replace “speed” with “rate”
Page 1, Line 29 – Insert “the” before “paper”
Page 1, Line 30 – Replace “of cotton” with “for cotton”
Page 1, Line 41 – Replace “proved” with “demonstrated”
Page 2, Lines 45-46 – Replace “a poor growth and less biomass” with “poor growth and reduced biomass”
Page 2, Line 46 – Replace “But excessive” with “Excessive” (do not start a sentence with But)
Page 2, Line 47 – Delete “the” before “canopy closure”
Page 2, Line 50 – Replace “But it” with “It”
Page 2, Line 65 – What does “0.3-1.6°C differences” mean?
Page 2, Line 85 (and other parts of the manuscript) – Replace “plant withdraw” with “plant senescence” or “plant death”
Page 3, Line 100 – What is “reproductive related parts (branch and branch leaves)” These sound like vegetative parts to me.
Page 3, Line 124 – Replace “figs” with “figures”
Page 3, Line 127 – Replace “Higher K fertilizers” with “Higher K fertilizer rates”
Page 3, Line 127 – Insert “higher” before “boll”
Page 4, Line 143 – Replace “cotton plants biomass” with “cotton plant biomass”
Page 4, Lines 142 and 143 – Replace “parts” with “part”
Page 4, Line 157 – Replace “continuously increasing” with “gradually diverged”
Page 4, Line 158 – Replace “RPB higher” with “RPB was higher”
Page 5, Line 178 – Replace “were appeared” with “occurred”
Page 5, Line 178 – Replace “stage that K3” with “stage where K3”
Page 6, Line 199 – Replace “while” with “end”
Page 6, Line 200 – Replace “among three” with “among the three”
Page 6, Line 202-203 – Replace “But VPB terminated FAP the” with “However, VPB terminated FAP on the”
Page 11, Line 281 – Replace “And the…” with “The…”
Page 12, Line 300 – Replace “differences were existed between” with “differences existed between the”
Page 12, Line 304 – Replace “owing” with “due”
Page 12, Line 305 – Replace “draught occurring” with “drought occurred”
Page 12, Line 308 – Replace “Similar result was observed in previous” with “A similar result was observed in a previous”
Page 12, Line 310 – Replace “K amount” with “K”
Page 12, Line 313 – Replace “boll weight in 2017” with “and in 2017”
Page 12, Line 316 – Replace “resulted a higher” with “resulting in a higher”
Page 12, Line 317 – Replace “induced” with “which reduced”
Page 12, Line 320 – Replace “enlarged” with “increased”
Page 12, Line 320 – Replace “resulting the significant” with “resulting in the significant”
Page 12, Lines 322-323 – Replace “in a sense of” with “in the context of”
Page 13, Line 329 – Replace “was benefited” with “benefited”
Page 13, Line 330 – Replace “Previous” with “A previous”
Page 13, Line 332 – Delete “was also reported”
Page 13, Lines 333-334 – Replace “other two K amounts during early growth period while” with “the other two K amounts during early growth period are”
Page 13, Line 335 – Replace “recorded higher” with “recorded at higher”
Page 13, Line 336 – Replace “That’s” with “This”
Page 13, Lines 340 & 350 – Replace “This result” with “These results”
Page 13, Line 334 – Replace “of” with “to”
Page 13, Line 349 – Replace “the similar behavioral” with “a similar behavior”.
Page 15, Line 448, Reference 33 – Replace “Ullah, N.; Brian, A.; Michael, B.; Daniel, T.” with “Najeeb, U.; Atwell, B.J.; Bange, M.P.; Tan, D.K.Y.”
Author Response
Response to Reviewer 2 Comments
Point 1: Title: It is not clear what "equal K" means. Do you mean K amounts equal to that of N application?
Response 1: I had followed your suggestion, and change the title to “Equal K application amounts to N achieved optimal yield biomass and better fiber quality of late sown cotton in Yangtze River Valley”.
Point 2: Page 1, Lines 13-14 - Replace "…a lowed N amounts (210 kg ha-1) under a condition of late sowing,…” with “… low N amounts (210 kg ha-1) under late sowing…”
Response 2: I had followed your suggestion, and replaced it.
Point 3: Page 1, Line 20 (and other parts of the manuscript) – Replace “speed” with “rate”
Response 3: I had followed your suggestion, and replaced it.
Point 4: Page 1, Line 29 – Insert “the” before “paper”
Response 4: I had followed your suggestion, and insert it.
Point 5: Page 1, Line 30 – Replace “of cotton” with “for cotton”
Response 5: I had followed your suggestion, and replaced it.
Point 6: Page 1, Line 41 – Replace “proved” with “demonstrated”
Response 6: I had followed your suggestion, and replaced it.
Point 7: Page 2, Lines 45-46 – Replace “a poor growth and less biomass” with “poor growth and reduced biomass”
Response 7: I had followed your suggestion, and replaced it.
Point 8: Page 2, Line 46 – Replace “But excessive” with “Excessive” (do not start a sentence with But)
Response 8: I had followed your suggestion, and replaced it.
Point 9: Page 2, Line 47 – Delete “the” before “canopy closure”
Response 9: I had followed your suggestion, and deleted it.
Point 10: Page 2, Line 50 – Replace “But it” with “It”
Response 10: I had followed your suggestion, and replaced it.
Point 11: Page 2, Line 65 – What does “0.3-1.6°C differences” mean?
Response 11: I had followed your suggestion, and replaced it with “air temperatures in 2016 were 0.3-1.6℃ lower than that in differences lower had been recorded from June to September between 2016 and 2017”.
Point 12: Page 2, Line 85 (and other parts of the manuscript) – Replace “plant withdraw” with “plant senescence” or “plant death”
Response 12: I had followed your suggestion, and replaced it.
Point 13: Page 3, Line 100 – What is “reproductive related parts (branch and branch leaves)” These sound like vegetative parts to me.
Response 13: I had followed your suggestion, and classified leaves, roots and stems to vegetative parts and squares, flowers and bolls to reproductive parts and make the same standard in both years.
Point 14: Page 3, Line 124 – Replace “figs” with “figures”
Response 14: I had followed your suggestion, and replaced it.
Point 15: Page 3, Line 127 – Replace “Higher K fertilizers” with “Higher K fertilizer rates”
Response 15: I had followed your suggestion, and replaced it.
Point 16: Page 3, Line 127 – Insert “higher” before “boll”
Response 16: I had followed your suggestion, and insert it.
Point 17: Page 4, Line 143 – Replace “cotton plants biomass” with “cotton plant biomass”
Response 17: I had followed your suggestion, and replaced it.
Point 18: Page 4, Lines 142 and 143 – Replace “parts” with “part”
Response 18: I had followed your suggestion, and replaced it.
Point 19: Page 4, Line 157 – Replace “continuously increasing” with “gradually diverged”
Response 19: I had followed your suggestion, and replaced it.
Point 20: Page 4, Line 158 – Replace “RPB higher” with “RPB was higher”
Response 20: I had followed your suggestion, and replaced it.
Point 21: Page 5, Line 178 – Replace “were appeared” with “occurred”
Response 21: I had followed your suggestion, and replaced it.
Point 22: Page 5, Line 178 – Replace “stage that K3” with “stage where K3”
Response 22: I had followed your suggestion, and replaced it.
Point 23: Page 6, Line 199 – Replace “while” with “end”
Response 23: I had followed your suggestion, and replaced it.
Point 24: Page 6, Line 200 – Replace “among three” with “among the three”
Response 24: I had followed your suggestion, and replaced it.
Point 25: Page 6, Line 202-203 – Replace “But VPB terminated FAP the” with “However, VPB terminated FAP on the”
Response 25: I had followed your suggestion, and replaced it.
Point 26: Page 11, Line 281 – Replace “And the…” with “The…”
Response 26: I had followed your suggestion, and replaced it.
Point 27: Page 12, Line 300 – Replace “differences were existed between” with “differences existed between the”
Response 27: I had followed your suggestion, and replaced it.
Point 28: Page 12, Line 304 – Replace “owing” with “due”
Response 28: I had followed your suggestion, and replaced it.
Point 29: Page 12, Line 305 – Replace “draught occurring” with “drought occurred”
Response 29: I had followed your suggestion, and replaced it.
Point 30: Page 12, Line 308 – Replace “Similar result was observed in previous” with “A similar result was observed in a previous”
Response 30: I had followed your suggestion, and replaced it.
Point 31: Page 12, Line 310 – Replace “K amount” with “K”
Response 31: I had followed your suggestion, and replaced it.
Point 32: Page 12, Line 313 – Replace “boll weight in 2017” with “and in 2017”
Response 32: I had followed your suggestion, and replaced it.
Point 33: Page 12, Line 316 – Replace “resulted a higher” with “resulting in a higher”
Response 33: I had followed your suggestion, and replaced it.
Point 34: Page 12, Line 317 – Replace “induced” with “which reduced”
Response 34: I had followed your suggestion, and replaced it.
Point 35: Page 12, Line 320 – Replace “enlarged” with “increased”
Response 35: I had followed your suggestion, and replaced it.
Point 36: Page 12, Line 320 – Replace “resulting the significant” with “resulting in the significant”
Response 36: I had followed your suggestion, and replaced it.
Point 37: Page 12, Lines 322-323 – Replace “in a sense of” with “in the context of”
Response 37: I had followed your suggestion, and replaced it.
Point 38: Page 13, Line 329 – Replace “was benefited” with “benefited”
Response 38: I had followed your suggestion, and replaced it.
Point 39: Page 13, Line 330 – Replace “Previous” with “A previous”
Response 39: I had followed your suggestion, and replaced it.
Point 40: Page 13, Line 332 – Delete “was also reported”
Response 40: I had followed your suggestion, and deleted it.
Point 41: Page 13, Lines 333-334 – Replace “other two K amounts during early growth period while” with “the other two K amounts during early growth period are”
Response 41: I had followed your suggestion, and replaced it.
Point 42: Page 13, Line 335 – Replace “recorded higher” with “recorded at higher”
Response 42: I had followed your suggestion, and replaced it.
Point 43: Page 13, Line 336 – Replace “That’s” with “This”
Response 43: I had followed your suggestion, and replaced it.
Point 44: Page 13, Lines 340 & 350 – Replace “This result” with “These results”
Response 44: I had followed your suggestion, and replaced it.
Point 45: Page 13, Line 334 – Replace “of” with “to”
Response 45: I had followed your suggestion, and replaced it.
Point 46: Page 13, Line 349 – Replace “the similar behavioral” with “a similar behavior”.
Response 46: I had followed your suggestion, and replaced it.
Point 47: Page 15, Line 448, Reference 33 – Replace “Ullah, N.; Brian, A.; Michael, B.; Daniel, T.” with “Najeeb, U.; Atwell, B.J.; Bange, M.P.; Tan, D.K.Y.”
Response 47: I had followed your suggestion, and replaced it.
Round 2
Reviewer 2 Report
The manuscript has been significantly improved and now warrants publication in Agronomy.